# QUESTION GENERATION USING A SCRATCHPAD ENCODER

## ABSTRACT

In this paper we introduce the *Scratchpad Encoder*, a novel addition to the sequence to sequence (seq2seq) framework and explore its effectiveness in generating natural language questions from a given logical form. The Scratchpad Encoder enables the decoder at each time step to modify all the encoder outputs, thus using the encoder as a "scratchpad" memory to keep track of what has been generated so far and to guide future generation. Experiments on a knowledge based question generation dataset show that our approach generates more fluent and expressive questions according to quantitative metrics and human judgments.

## 1 INTRODUCTION

Given the data driven nature of today's question answering (QA) systems, the size and quality of training data play a major role in a system's ability to answer questions correctly. In a knowledge-based question answering system, where the input is a natural language question and the answer generated is retrieved from the knowledge base (KB), the standard approach is to *parse* the question into a logical form that can query the knowledge base. (See Fig. 1 for an example.) This intermediate logical form could be Lambda-DCS (Liang (2013)), SPARQL (Harris et al. (2013)), or any other form interpretable by the knowledge base's query engine.

One of the main challenges in training robust knowledge-based QA models is acquiring a large amount of diverse labeled data. Currently, the largest publicly available datasets for knowledge-based QA are on the order of 5-6 thousand queries [1] (tau Yih et al. (2016); Berant et al. (2013); Su et al. (2016)), which is relatively small when compared with reading comprehension datasets like SQuAD (Rajpurkar et al. (2016)) that have on the order of 100,000 examples, or machine translation datasets containing millions to tens of millions of parallel sentences.

Several complications arise when constructing labeled corpora for knowledge-based QA. Namely, since non-expert crowd workers are not familiar with logical form languages, collecting *(query, logical form)* pairs can be a difficult and slow process (Liang et al. (2016); Reddy et al. (2014)) [2]. One could bypass crowd sourcing logical forms by training a model on *(question, answer)* pairs, treating the intermediate logical as a *latent variable* (Berant et al. (2013); Kwiatkowski et al. (2013); Yao & Van Durme (2014). Unfortunately, collecting *only* (query, answer) pairs is also difficult. To create the WebQuestions dataset Berant et al. (2013), judges were given 100,000 questions and asked to find the answers in Freebase. However, in only ~6,000 questions of these 100,000 (6%) were two judges able to arrive at the same answer.

In order to collect arbitrarily large datasets, we need methods requiring less human intervention and expertise. In this paper we explore the feasibility of generating meaningful *questions* given a logical form (i.e. SPARQL). The intuition is that it is easier to have a programmer with domain expertise generate a large set of programs, rather than train an average human judge to master SPARQL.

---

[1] The SimpleQuestions dataset (Bordes et al. (2015)) contains 100,000 examples, but cannot be considered as a general knowledge-base QA or semantic parsing dataset because all the questions have one logical form only: "*Select Entity.Attribute from KB*". Here the parser's job is to only fill out the values of *Entity* and *Attribute*. Our goal is to be able to answer more complex questions that require logical forms beyond those that can be described as only triple selection (See Fig 1).

[2] WebQuestionsSP tau Yih et al. (2016) dataset used crowdsourcing to collect (query, logical form) pairs, but the judges were familiar with Freebase, and a special interface was created for them

| SPARQL: | SELECT DISTINCT ? x
WHERE {
 FILTER (?x != ns:m.078ffw)
 FILTER (!isLiteral(?x) OR lang(?x) = '' OR langMatches(lang(?x), 'en'))
 ns:m.078ffw ns:book.literary_series.works_in_this_series ?x .
 ?x ns:book.written_work.date_of_first_publication ?sk0 .
}
ORDER BY xsd:datetime(?sk0)
LIMIT 1 |
| --- | --- |
| Question: | What is the name of the first harry potter novel? |

Figure 1: A $(logical\ form, question)$ pair from WebQuestionsSP (tau Yih et al. (2016)).

Recent work has even shown question generation from a logical form leads to improvements in semantic parsing (Guo et al. (2018)). We adapt the sequence to sequence (seq2seq) framework (Sutskever et al. (2014)) given its success in tasks such as machine translation (Bahdanau et al. (2014)) and traditional semantic parsing (Dong & Lapata (2016)). Despite their success, seq2seq with attention models fail to produce fluent output to the level of specificity and quality necessary for our task. Furthermore, they often fail to keep track of what has already been generated by the decoder, or copied from the input tokens in the case of *Copynet* (He et al. (2017) (see Table 7) often leading to erroneous repetitions. Our approach is aimed at mitigating these issues.

We introduce a novel write mechanism to the seq2seq framework that significantly outperforms several baselines termed the *Scratchpad Encoder*. Put simply, we allow the decoder to keep notes as it decodes; we keep the standard seq2seq encoder (Sutskever et al. (2014)), but at each time step allow the decoder to attentively write to the encoder outputs. In this way, we use the encoder as a "scratchpad" to direct future generation. In both quantitative evaluations of the generated questions and human judgments of their fluency and adequacy, our model attains significant improvement over standard seq2seq, *Copynet* (Gu et al. (2016)), and a coverage-enhanced approaches Tu et al. (2016); See et al. (2017). Furthermore, human judges strongly prefer questions generated using the Scratchpad Encoder over those produced using *Copynet* and a coverage-enhanced approaches — $89.44\%$ and $61.36\%$ of the time, respectively.

The contributions of this work are two-fold:

1. We introduce Scratchpad Encoder, a novel enhancement to the seq2seq framework allowing the decoder to see what has been generated thus far, and outperforming multiple baselines on quantitative and qualitative metrics.

2. We use Scratchpad Encoder to automatically generate questions given a corresponding SPARQL query, making it possible to generate a large high quality dataset of knowledge base (query, logical form) pairs.

## 2 MODEL BASICS/BACKGROUND

We build off of a standard seq2seq setup with (a) an *encoder* operating off the *logical form*, (b) a *decoder* outputting the *generated question*, and (c) a task-specific *attention mechanism* so that the decoder may focus on different parts of the logical form as it decodes. We describe these components below before detailing the *copy mechanism* and Scratchpad Encoder.

**Encoder** Let the input sequence (tokens of the *logical form*) of length $T$ be indexed by the subscript $t$, so that the sequence of word vectors input into the model is $x_0...x_T = x_{0..T}$. We encode the input via a bidirectional 2 layer GRU (Cho et al. (2014)), resulting in encoder outputs $h_{0..T}$.

**Decoder** Let the decoding sequence (tokens of the *question*) be indexed by the superscript $i$ such that $s^i$ is the state of the decoder at decoding timestep $i$. The decoder's initial state $s^0$ is set to the final output of the encoder in both directions. The decoder uses a recurrence new_state = $f^{(2)}$(current _state, input), where we use $f^{(1)}$ to refer to an $l$-layer GRU from now on.

**Attention** At every decoding step i, the decoder computes an attentive read ($\text{attn}_{\text{read}}^i$) over the encoder states ($h_{0..T}$) (as in Bahdanau et al. (2014)). This is computed by first computing a *score* for each encoder output ($h_t$) via an MLP as follows. (We denote the concatenation of vectors $z_0, ..., z_n$ as $[z_0; ...; z_n]$.)

$$\text{score}_t^i = W^1(W^2[s^i; h_t]^T) \tag{1}$$

where $W^1$ and $W^2$ are learned parameters. These scores ($\text{score}_{0..T}^i$) are then normalized into a probability distribution and used as the weights to compute a weighted average of encoder outputs (termed the *attentive read*), allowing the decoder to focus on different parts of the input at different timesteps i:

$$a_{0..T}^i = \text{softmax}(\text{score}_{0..T}^i) \tag{2}$$

$$\text{attn}_{\text{read}}^i = \sum_{t=0}^{T}(a_t^i * h_t) \tag{3}$$

**Update** The decoder then computes its new state using the word vector $\hat{y}^{i-1}$ for the previous output $y^{i-1}$ along with the attentive read ($\text{attn}_{\text{read}}^i$) to obtain the post-read state ($s_{\text{post\_read}}^i$):

$$s^{i+1} = s_{\text{post\_read}}^i = f^{(2)}(s^i, [\hat{y}^{i-1}; \text{attn}_{\text{read}}^i]) \tag{4}$$

**Generation** At every step, the decoder computes a distribution over output tokens from the post-read state ($s_{\text{post\_read}}^i$):

$$y^i = \text{softmax}(W^{\text{out}} s_{\text{post\_read}}^i) \tag{5}$$

where $W^{\text{out}}$ is a learned parameter.

**Cross-Vocabulary Copying** We noticed that many tokens that appear in the logical form are also present in the natural language form for each example. In fact, nearly half of the tokens in the question appear in the corresponding SPARQL of the WebQuestionSP dataset (tau Yih et al. (2016)) Hence, we give the network the ability to copy (Gu et al. (2016)) from its input by redefining the output distribution $y^i$ as a mixture between the *generation distribution* described previously (Eq. 5) and a *copy distribution* (Eq. 7).

$$y^i = p_{\text{gen}}^i * \text{softmax}(W^{\text{out}} s_{\text{post\_read}}^i) + (1 - p_{\text{gen}}^i) * \text{copy}(s_{\text{post\_read}}^i, h_{0..T}, x_{0..T}) \tag{6}$$

where copy is a distribution over the vocabulary V. We only consider "copyable" tokens ($V_c$) that appear in both the logical form input vocabulary ($V_i$) and question output vocabulary ($V_o$) ($V_c = V_i \cap V_o$). We define copy as:

$$\text{copy}(s_{\text{post\_read}}^i, h_{0..T}, x_{0..T})_v = \sum_{t=0}^{T}(c_t^i * 1_{x_t == v}) \tag{7}$$

The copy-attention distribution $c_{0..T}^i$ is computed in a similar fashion to the attentive-read distribution ($a_{0..T}^i$) computed earlier (Eq. 3), except we use the post-read decoder state ($s_{\text{post\_read}}^i$) so the network can take into account the attentive read ($\text{attn}_{\text{read}}^i$) as well as the previously generated token ($\hat{y}^{i-1}$) when computing $\text{copy\_score}_t^i$.

$$\text{copy\_score}_t^i = W^1(W^2[s_{\text{post\_read}}^i; h_t]^T)) \tag{8}$$

$$c_{0..T}^i = \text{softmax}(\text{copy\_score}_{0..T}^i) \tag{9}$$

$$\text{attn}_{\text{copy}}^i = \sum_{t=0}^{T}(c_t^i * h_t) \tag{10}$$

where $W^1$ and $W^2$ are learned parameters with the same dimensionalities as in the previous case.

Finally, we need to compute "how much" we are generating vs. copying ($p_{\text{gen}}^i$), again using the post-read decoder state ($s_{\text{post\_read}}^i$) and the copy read ($\text{attn}_{\text{copy}}^i$) so the network can take the attentive read ($\text{attn}_{\text{read}}^i$) and previously generated token ($\hat{y}^{i-1}$) into account.

$$p_{\text{gen}}^i = \sigma(W^1 \text{ReLU}(W^2[s_{\text{post\_read}}^i; \text{attn}_{\text{copy}}^i]^T)) \tag{11}$$

## 3 KEEPING NOTES WITH A SCRATCHPAD ENCODER

Although modern seq2seq models (including common extensions like *Copynet* tend to do well on multiple tasks and have led to promising improvements across the board (Bahdanau et al. (2014); Sutskever et al. (2014); Dong & Lapata (2016)), they often have issues with over and under-generation, particularly with regards to repetition or copying (See et al. (2017); Tu et al. (2016), as well as integrating new information (See Eric & Manning (2017)). We propose a method that allows the network to update its encoding of the input at every step of decoding. Intuitively, we add one simple step to the decoder: treat the encoder states as a *scratchpad*, writing to it as if it were an external memory.

Up until now, the decoder's workflow at every step i is as follows:

1. *Reads* attentively from the encoder outputs ($\mathrm{attn}^i_{\mathrm{read}}$)  (See Eq. 3)
2. *Updates* its state ($s^i_{\mathrm{post\_read}}$)  (See Eq. 4)
3. *Outputs* a distribution ($y^i$) over the output vocabulary  (See Eq. 6)

With the *Scratchpad Encoder* we add a fourth step:

4. *Write* an update ($u^i$) to the encoder states ($h_{0..T}$) in an attentive fashion ($\alpha^i_{0..T}$) using the post-read decoder state ($s^i_{\mathrm{post\_read}}$), treating the encoder states ($h_{0..T}$) as if they were cells in an external memory:

$$h^{i+1}_t = \alpha^i_t h^i_t + (1 - \alpha^i_t)u^i \tag{12}$$

$$\alpha^i_t = \sigma(W^1(W^2[s^i_{\mathrm{post\_read}}; \mathrm{attn}^i_{\mathrm{copy}}; h^i_t]^T)) \tag{13}$$

$$u^i = \mathrm{Tanh}(W^3\mathrm{ReLU}(W^4[s^i_{\mathrm{post\_read}}; \mathrm{attn}^i_{\mathrm{copy}}]^T)) \tag{14}$$

Tanh is used to ensure that $h^{i+1}_t$ remains in the range $[-1, 1]$, since $h^i_t \in [-1, 1]$ as $h^0_t$ is the output of a GRU. $\alpha^i_t$ can be understood as an update gate for the representation of the input sequence $h_{0..T}$, signifying how much to overwrite a cell versus keeping past information.

While decoding, it is advantageous for the network to keep track of which tokens have been generated and which locations have been attended. By allowing the decoder to write to the encoder we can easily track this information. By keeping the information outside of the decoder GRU we also preserve capacity in the decoder for other subtasks like smoothing.

The Scratchpad Encoder is independent of the copy mechanism that we present here, meaning it can be an addition to any seq2seq with attention framework. To use the Scratchpad Encoder without the copy mechanism replace $\mathrm{attn}^i_{\mathrm{copy}}$ with $\mathrm{attn}^i_{\mathrm{read}}$ in the equations above.

## 4 PREPROCESSING AND TRAINING

**Preprocessing**   We split on special characters and camelCasing (see Fig. 2). Since many of these strings are compositional, tokenizing in this fashion allows the network to take advantage of this fact. This results in a large variance in sequence lengths (min 20, max 338), with an average sequence length of $80.84$ tokens.

**Training**   All models were trained for 75 epochs with a batch size of 32, a hidden size of 512, and a word vector size of 300. Dropout is used on every layer of all GRUs except the output layer, with a drop probability of $0.5$. Where Glove vectors (Pennington et al. (2014)) are used to initialize word vectors, we use 300-dimensional vectors trained on Wikipedia and Gigaword ($6B.300D$). The Adam optimizer (Kingma & Ba (2014)) was used, with a learning rate of $1e^{-4}$ and a teacher forcing (Williams & Zipser (1989) probability of $0.5$. These hyperparameters were tuned for our Seq2Seq baselines and held constant for the rest of the models (Copynet, Coverage, Scratchpad). The vocabulary consists of all tokens appearing at least once in the training set.

```
SELECT DISTINCT ?x
WHERE {
FILTER (?x != ns:m.01bkb)
FILTER (!isLiteral(?x) OR lang(?x) = '' OR langMatches(lang(?x), 'en'))
ns:m.01bkb ns:travel.travel_destination.how_to_get_here ?y .
?y ns:travel.transportation.transport_terminus ?x .
?y ns:travel.transportation.mode_of_transportation ns:m.03qb78c .
}
```
(Raw)

```
select distinct ? x
where {
filter ( ? x ! = ent: bali )
filter ( ! is literal ( ? x ) or lang ( ? x ) = ' ' or lang matches ( lang ( ? x ) , ' en ' ) )
ent: bali ns travel . travel _ destination . how _ to _ get _ here ? y .
? y ns travel . transportation . transport _ terminus ? x . ? y ns travel . transportation . mode _ of _ transportation ent: air transportation .
}
```
(Preprocessed)

Figure 2: To preprocess SPARQL, we split on: special characters (? ! = : . _ ( ) { } '), ':' not followed by '/', and camelCasing. Entity IDs are replaced with the 'ent:' token followed by the entity's full name. We then lowercase all strings.

## 5 EXPERIMENTAL RESULTS

### 5.1 DATA

We use a standard dataset for semantic parsing, WebQuestionsSP (tau Yih et al. (2016)), which consists of *(question, logical form)* pairs where the logical form is in SPARQL. The dataset contains 3098 training examples, with additional 1639 for testing. All the following results are reported for the test fold.

In addition to knowledge base question generation, we evaluate our approach on the task of generating questions from SQL statements. For this we use the WikiSQL dataset (Zhong et al. (2017)), and report the results in Appendix A.

### 5.2 BASELINES

We compare our Scratchpad Encoder against 4 baselines: (1) Seq2Seq, (2) Seq2Seq with Priors, (3) *Copynet,* and (4) *Coverage* which is a method from machine translation that aims to solve attention-based coverage problems (Tu et al. (2016)). Seq2Seq is the standard approach introduced in (Sutskever et al. (2014)), whereas "Seq2Seq + Priors" has word vectors initialized from glove embeddings, uses beam search, and uses smarter preprocessing (replace entity IDs with full entity names). *Copynet* (He et al. (2017)) baseline gives the Seq2Seq model the ability to copy vocabulary from the source to the target, as detailed in equation 7. Table 1 provides a comparison of all the baselines.

**Coverage Baseline** Our final baseline is a copy-enhanced seq2seq model with attention with a neural "coverage" mechanism added as in Tu et al. (2016). It was originally introduced for the task of machine translation to address the problems of over and under translation. To do so, they add a "coverage vector" ($\text{cov}_t^i$) for each position ($h_t$) in the encoder outputs, to *keep track* of the history of (copy) attentions ($c_t^0..c_t^{i-1}$) so far.:

$$\text{cov}_t^i = f^{(1)}(\text{cov}_t^{i-1}, [s_{\text{post\_read}}^i; h_t; c_t^i]) \tag{15}$$

This requires adding an additional RNN to the model ($f^{(1)}$). Each coverage vector ($\text{cov}_t^i$) is now taken into account when calculating the distribution for copying ($c_t^i$, Eq. 9), which also affects the calculation of the copy-read ($\text{attn}_{\text{copy}}^i$, Eq. 10). This is done by modifying the calculation of copy_score$_t^i$ from Eq. 8 to:

$$\text{copy\_score}_t^i = W^1(W^2[s_{\text{post\_read}}^i; h_t; \text{cov}_t^{i-1}]^T) \tag{16}$$

For a fair comparison, we compare against a coverage vector of size 10, the largest used in Tu et al. (2016).

| Name | Replace Entity ID | Glove Word Embeddings | Beam Search | Ability to Copy | Coverage RNN | Scratchpad Encoder |
|---|---|---|---|---|---|---|
| Seq2Seq | ✗ | ✗ | ✗ | ✗ | ✗ | ✗ |
| + Priors | ✓ | ✓ | Beam Size 2 | ✗ | ✗ | ✗ |
| Copynet | ✓ | ✓ | ✗ | ✓ | ✗ | ✗ |
| + Coverage | ✓ | ✓ | ✗ | ✓ | ✓ | ✗ |
| + Scratchpad | ✓ | ✓ | ✗ | ✓ | ✗ | ✓ |

Table 1: To validate the method, we compare against a pure *seq2seq with attention* baseline, along with a tuned ("+ Priors") version and a *Copynet* with the same improvements except for beam search. We then enhance a copynet with a *coverage mechanism* and a *scratchpad encoder*.

## 5.3 QUANTITATIVE EVALUATIONS

In order to quantitatively evaluate the performance of methods, we compute BLEU (for a precision-based metric), ROUGE-LCS (for a recall-based metric), and METEOR (to deal with stemming and synonyms). We run these metrics at both a corpus level (i.e. how natural are output questions), and at a per-sentence level (i.e. how well do output questions exactly *match* the gold question). We evaluate on examples in the test set that do not contain out of vocabulary tokens. Table 2 shows the performance of each baseline on all three metrics. From the table it is clear that our approach, Scratchpad Encoder, outperforms all baselines on all the metrics.

| Model | Per-Sentence | | | Corpus-Level | | |
|---|---|---|---|---|---|---|
| | Bleu | Meteor | Rouge-L | Bleu | Meteor | Rouge-L |
| Baseline | 6.1 | 21.5 | 42.5 | 15.11 | 20.79 | 42.48 |
| + Priors | 7.51 | 23.9 | 47.1 | 17.96 | 22.9 | 47.13 |
| Copynet | 6.89 | 27.1 | 52.5 | 17.42 | 26.03 | 52.56 |
| + Coverage | 14.55 | 33.7 | 58.9 | 26.78 | 30.86 | 58.91 |
| + Scratchpad | **15.29** | **34.7** | **59.5** | **27.64** | **31.49** | **59.44** |

Table 2: Methods allowing the model to keep track of past attention (*Coverage*, *Scratchpad*) significantly improve performance when combined with a copy mechanism. The *Scratchpad Encoder* achieves the best performance.

## 5.4 HUMAN EVALUATION

Although quantitative metrics such as BLEU tend to correlate with human judgments for machine translation tasks (Bojar et al. (2017)), they do not always correlate with the human assessed quality of generated text for all tasks (Stent et al. (2005); Liu et al. (2016)). We use two standard human evaluation metrics from the machine translation community: (1) *Adequacy*, and (2) *Fluency* (Bojar et al. (2017)). In computing the adequacy metric, human judges are presented with a reference translation and the system proposed translation, and are asked to rate the adequacy of the proposed translation in conveying the meaning of the reference translation on a scale from 0-10. For fluency, the judges are asked to rate, on a scale from 0-10, whether the proposed translation is a fluent English sentence. Table 4 summarizes the human evaluation results for our Scratchpad Encoder and two more baselines. As the table shows, the judges assigned higher fluency and adequacy scores to our approach than the coverage based decoder and the copynet one. In the table we also report the fluency score of the gold questions as a way to measure the gap between the generate questions and

| Expected | Copynet | Coverage | Scratchpad |
|---|---|---|---|
| what was thomas jefferson role in the declaration of independence? | what is jefferson **jefferson** famous for? | what was thomas jefferson famous for? | what jobs did thomas jefferson have? |
| who does chris hemsworth have a baby with? | what does chris **chris chris** with? | who is chris **chris** hemsworth? | who is chris hemsworth married to? |
| what type of music did vivaldi compose? | what type of music did vivaldi **vivaldi** sing? | what music did antonio vivaldi compose? | what music did antonio vivaldi compose? |
| what else did ben franklin invent? | what is the inventions **inventions** of franklin **franklin**? | what was the inventions benjamin **benjamin** franklin's children? | what are some inventions benjamin franklin made? |

Table 3: Example instances from the test set of the copy-mechanism of *copynet* over-firing where the *scratchpad* model copies correctly.

| Model | Fluency | Adequacy |
|---|---|---|
| Gold | 9.13 | ✗ |
| Copynet | 5.18 | 5.23 |
| + Coverage | 6.64 | 6.16 |
| + Scratchpad | **7.38** | **6.59** |

Table 4: Human evaluations show that the *Scratchpad Encoder* delivers a large improvement in both *fluency* and *adequacy* over *Copynet* and *Coverage*.

| Scratchpad vs. Copynet | | Scratchpad vs. Coverage | |
|---|---|---|---|
| Both Good | 9.26% | Both Good | 15.11% |
| Scratchpad | 37.78% | Scratchpad | 23.80% |
| Copynet | 6.46% | Coverage | 14.99% |
| Both Bad | 46.5% | Both Bad | 43.07% |
| Win Rate | 89.44% | Win Rate | 61.36% |

Table 5: The percentage of times judges preferred one result over the other. In a Head-to-Head evaluation the output of *Scratchpad Encoder* is 9 and 2 times as likely to be chosen vs. *Copynet* and *Coverage*, respectively. Win rate is the percentage of times Scratchpad was picked when the judges chose a single winner (not a tie).

the expected ones. Our approach is nearly 2 full points behind the gold when it comes to generation fluency.

In addition to adequacy and fluency, we design a side-by-side experiment to find out which approach generates better questions in pairwise comparison fashion. In the study, judges are presented with 2 generated questions from 2 different systems, along with the *reference* question and are asked which of the two systems presents a better *paraphrase* to the reference question. The judges took into consideration the grammatical correctness of the question and its ability to capture the meaning of the reference question fluently. Table 5 shows the result of running the head-to-head evaluation between scratchpad output and both copynet and coverage baselines. As the table shows, human judges are four times as likely to prefer scratchpad generated questions over copynet, and nearly two times over coverage. Table 7 shows examples of generated questions by the different approaches.

## 6 RELATED WORK

In the question generation domain, there has been a recent surge in research on generating questions for a given paragraph of text (Song et al. (2017); Du et al. (2017); Tang et al. (2017); Duan et al. (2017); Wang et al. (2018); Yao et al. (2018)). This work focuses on applications for machine reading comprehension where a paragraph coupled with a snippet containing the answer are used to generate questions. This approach arguably contains more contextual information to help guide the

model than using only a logical form, sometimes syntactic manipulation of the answer paragraph alone is sufficient to generate a question. In general, most of the work in this area has been a variant of the seq2seq approach. In Song et al. (2017), a seq2seq model with copynet and a coverage mechanism (Tu et al. (2016)) is used to achieve state-of-the-art results. We have demonstrated that our Scratchpad Encoder outperforms this approach in both quantitative and qualitative evaluations.

In the knowledge based question generation domain, early work on translating SPARQL queries into natural language focused on generating a human readable description of SPARQL queries to guide query writers (Ngonga Ngomo et al. (2013a;b))). However, that approach relied on hand-crafted rules to translate certain words appearing in the SPARQL query. Later work on automatically generating questions from SPARQL queries have also relied on manually crafted templates to map selected categories of SPARQL queries to questions (Trivedi et al. (2017); Seyler et al. (2017)). In Serban et al. (2016) knowledge base triplets are used to generate questions using encoder-decoder framework that operates on entity and predicate embeddings trained using TransE (Bordes et al. (2011)). Later, Elsahar et al. (2018) extended this approach to support unseen predicates. Both approaches operate on triplets, meaning they have limited capability beyond generating simple questions. Since our approach operates on the more expressive SPARQL query (logical form) we can produce far more complex questions.

Summarizing source code is another area where we draw inspiration (Iyer et al. (2016)). Approaches in this area are largely based on attentive seq2seq models, although an Abstract Syntax Tree aware encoder was introduced recently (Alon et al. (2018)). Code summaries tend to be more descriptive than our factoidal question generation. In the dataset provided by (Iyer et al. (2016)) the examples were collected from StackOverFlow, where the questions are generally about how to fix a piece of code. SQL queries bear a striking resemblance to SPARQL, so we test approach on the Zhong et al. (2017) dataset, where we to generate questions from SQL statements. The experiments presented in Appendix A, demonstrate the same performance gains obtained on the WebQuestionsSP dataset, outperforming all baselines.

Closest to our work, in the general paradigm of seq2seq learning, is the coverage mechanism introduced in Tu et. al Tu et al. (2016) and later adapted for summarization in See et al. (2017). Both works try to minimize erroneous repetitions generated by a copy mechanism by introducing a new vector to keep track of what has been used from the encoder thus far. In Tu et al. (2016), for example, use an extra GRU to keep track of this information, whereas See et al. (2017) keeps track of the sum of attention weights and adds a penalty to the loss function based on it to discourage repetition. Our approach is much simpler than either solution since it does not require any extra vectors or an additional loss term; rather, the encoder vector itself is being used as *scratch memory*. Our experiments also show that for the question generation task, the Scratchpad Encoder performs better than coverage based approaches.

Our idea was inspired by the dialogue generation work of Eric & Manning (2017) in which the entire sequence of interactions is re-encoded every time a response is generated by the decoder. This has also been explored in Elbayad et al. (2018) to great effect. Unfortunately, all of these methods have an $O(n^2)$ runtime, which scales poorly with long sequences and large input examples. In addition, vast amounts of memory are required during training to implement Eric & Manning (2017) or Elbayad et al. (2018)). Whereas our module is $O(n)$ in runtime, and does not add memory beyond a constant factor. Finally, our work is very similar to the research done on using external memories for generation (e.g., Bordes et al. (2016); Eric et al. (2017)) and could be viewed as more efficient way to initialize such an external memory.

# 7 Conclusion

In this paper, we addressed the task of generating factoidal questions from a knowledge base using SPARQL queries as input. We introduced a novel write operator to the seq2seq framework, which we call a Scratchpad Encoder. The Scratchpad Encoder helps the decoder to keep track of what tokens have been generated so far, and guide future generation. It outperforms multiple baselines including Seq2Seq, Copynet, and *Coverage*, in both quantitative evaluations and in human judgments Our module is conceptually simple and easy to add to any seq2seq model with attention.

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

## APPENDIX A    GENERATING QUESTIONS FROM SQL

To demonstrate the generalizable nature of this approach we also evaluate the Scratchpad Encoder on the WikiSQL dataset Zhong et al. (2017), where our task becomes generating natural language questions from SQL statements. We perform the same preprocessing applied to the WebQuestionsSP dataset, and train along with the same baselines from before (Seq2Seq with priors, Copynet, and Coverage). The quantitative results are given in Table 6. From the table it is clear that our approach outperforms all baselines on all metrics. Furthermore, the improvement is consistent with the performance gains obtained on the WebQuestionsSP, indicating the efficacy of this approach across logical form formats.

| Model | Per-Sentence | | | Corpus-Level | | |
|---|---|---|---|---|---|---|
| | Bleu | Meteor | Rouge-L | Bleu | Meteor | Rouge-L |
| Baseline | 9.94 | 26.71 | 47.96 | 17.34 | 25.34 | 47.96 |
| Copynet | 8.04 | 24.66 | 46.82 | 15.11 | 23.53 | 46.82 |
| + Coverage | 15.76 | 34.04 | 54.94 | 25.01 | 32.38 | 54.94 |
| + Scratchpad | **16.89** | **34.47** | **55.69** | **26.10** | **32.76** | **55.69** |

Table 6: Methods allowing the model to keep track of past attention (*Coverage*, *Scratchpad*) significantly improve performance when combined with a copy mechanism. The *Scratchpad Encoder* achieves the best performance.

| Expected | Copynet | Coverage | Scratchpad |
|---|---|---|---|
| what is the total number of attendance ( s ) , when away is real juventud ? | what was the attendance for juventud **juventud** ? | what was the attendance against real juventud ? | what was the total attendance against real juventud |
| name the pictorials when the interview subject is steve jobs | in the issue in which the steve were **were** steve **steve** , what were the steve ? | in the issue in **in in** the issue where jobs **jobs** was the interview subject | which were the pictorials in which the interview subject was steve jobs ? |
| name the % of popular vote for election for 1926 | what is the percentage of the vote 1926 in **1926** ? | what is the percentage of the vote in the candidate of 1926 ? | what was the percentage of popular vote for the 1926 election ? |
| what is the lowest rank ? | what is the smallest rank ? a **rank ?** | what is the smallest rank for a **rank** ? | what 's the minimal rank of a athlete shown in the chart ? |

Table 7: Example instances from the test set of the copy-mechanism of *copynet* over-firing where the *scratchpad* model copies correctly.

## APPENDIX B    ANALYSIS OF OVERCOPYING

As our approach was introduced to help the decoder keep track of what has been generated (and copied) so far, we introduce a metric for *over copying* which is defined as the difference between the number of times a word type ($w$) is present in reference text ($r$) compared to the generated text ($g$)

$$\text{OverCopy}(\text{corpus}) = \frac{\sum \text{ExampleOverCopy}(r, g)}{\text{Size}(\text{corpus})} \qquad (17)$$

$$\text{ExampleOverCopy}[r, g] = \sum_w \max(\text{delta}[w], 0) \qquad (18)$$

$$\text{delta}[w] = \text{Count}_p[w] - \text{Count}_g[w] \qquad (19)$$

On the WebQuestionsSP dataset, our approach achieves an average corpus wide over copying score of 0.23, whereas coverage and copynet achieve 0.32 and 0.66, respectively. Our approach reduces over copying by $28\%$ over the next best approach.

Recall that copying was found very useful for question generation since a question typically shares the same vocabulary with the logical form. Table 8 shows examples highlighting the shared tokens between questions and logical forms.

what are the **official** languages in **spain** ?
what is the first **book sherlock holmes appeared** in ?
when did the **new york knicks** win a **championship** ?
who portrayed **indiana jones** in **raiders of the lost ark** ?
from what **university** did **president obama** receive his **bachelor** 's **degree**

Table 8: Example questions from the training set of WebQuestionsSP dataset, with tokens that also appear in the sparql in bold. Nearly half ($47.36\%$) of tokens in questions appear in the corresponding sparql, suggesting that the ability to copy can improve performance on this task.

## APPENDIX C    MODEL DIAGRAM

The following diagram outlines the interactions between the different components of the system. Boxes are annotated with the corresponding equation where applicable.

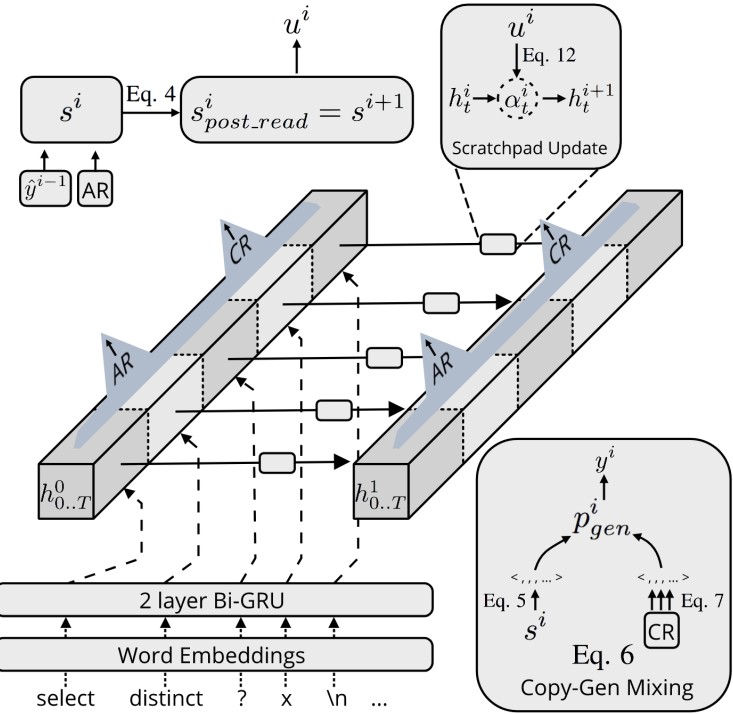

Figure 3: Model diagram demonstrating a copynet enhanced with a scratchpad encoder. The attentive read $attn^i_{read}$ and copy read $attn^i_{copy}$ are denoted AR and CR, respectively. $h^0_{0..T}$ is the output of the encoder, and $h^1_{0..T}$ the new values of those states after the scratchpad update.

