# OpenReview forum: "Question Generation using a Scratchpad Encoder"
_ICLR.cc/2019/Conference_

### Official Review · AnonReviewer1 · 2018-10-29
**Generating questions is an interesting task, but it is a kind of natural language generation task and the paper does not consider and they have proposed very similar ideas already**

**Rating:** 4
**Confidence:** 5

**Review:**

The paper studies the problem of question generation from sparql queries. The motivation is to generate more training data for knowledge base question answering systems to be trained on. However, this task is an instance of natural language generation: given a meaning representation (quite often a database record), generate the natural language text correspoding to it. And previous work on this topic has proposed very similar ideas to the scratchpad proposed here in order to keep track of what the neural decoder has already generated, here are two of them:
- Semantically Conditioned LSTM-based Natural Language Generation for Spoken Dialogue Systems
Tsung-Hsien Wen, Milica Gasic, Nikola Mrksic, Pei-Hao Su, David Vandyke, Steve Young, EMNLP 2015: https://arxiv.org/abs/1508.01745
- Globally Coherent Text Generation with Neural Checklist Models
Chloe Kiddon Luke Zettlemoyer Yejin Choi: https://aclweb.org/anthology/D16-1032
Thus the main novelty claim of the paper needs to be hedged appropriately. Also, to demonstrate the superiority of the proposed method an appropriate comparison against previous work is needed.

Some other points:
- How is the linearization of the inout done? It  typically matters
- Given the small size of the dataset, I would propose experimenting with non-neural approaches as well, which are also quite common in NLG.
- On the human evaluation: showing the gold standard reference to the judges introduces bias to the evaluation which is inappropriate as in language generation tasks there are multiple correct answers. See this paper for discussion in the context of machine translation: http://www.aclweb.org/anthology/P16-2013
- For the automatic evaluation measures there should be multiple references per SPARQL query since this is how BLEU et al are supposed to be used. Also, this would allow to compare the references against each other (filling in the missing number in Table 4) and this would allow an evaluation of the evaluation itself: while perfect scores are unlikely, the human references should be much better than the systems.
- In the outputs shown in Table 3, the questions generated by the scratchpad encoder often seem to be too general compared to the gold standard, or incorrect. E.g. "what job did jefferson have" is semntically related to his role in the declaration of independence but rather different. SImilarly, being married to someone is not the same as having a baby with someone. While I could imagine human judges preferring them as they are fluent, I think they are wrong as they express a different meaning than the SPARQL query they are supposed to express. What were the guidelines used?

---

> ### Author Response · Authors · 2018-12-11
> **We provide a conceptually simple method delivering state of the art performance by significantly outperforming standard methods for natural language generation.**
>
> We are aware of the related work you mention. Please note that unfortunately the “Semantically Conditioned LSTM…” is not directly comparable because, as they state in their paper, “the generator is further conditioned on a control vector d, a 1-hot representation of the dialogue act (DA) type and its slot-value pairs”. Our goal is to work with arbitrarily complex questions that map to correspondingly arbitrarily complex logical forms and not a very restricted set of logical forms that could be represented in a one-hot fashion.
> Please do note that we ran 2 sets of human evaluations (Adequacy and Fluency), as is standard in Machine translation in order to deal with the evaluation bias problem you describe - we took this into account when conducting experiments and will make it more clear in a revised version. We also observe significant improvements in both human evaluations, suggesting that the improvement comes from our method and not from evaluation bias.
> Our dataset only contains a single logical form for each question and vice-versa, making it impossible to evaluate quantitative metrics (bleu, rouge, meteor) in the multi-reference setting you describe. Please also note that metrics like bleu and rouge have been commonly used in a non multi-reference setting by significant work in the natural language processing community.
> We thank the reviewer for their comments and will take them into account in a revised version.

---

### Official Review · AnonReviewer2 · 2018-11-03

**Rating:** 3
**Confidence:** 5

**Review:**



This paper tackles the question generation problem from a logical form and proposes an addition called Scratchpad Encoder to the standard seq2seq framework. The new model has been tested on the WebQuestionsSP and the WikiSQL datasets, with both automatic and human evaluation, compared to the baselines with copy and coverage mechanisms.

Major points:

Overall, I think this paper is not good enough for an ICLR paper and the presentation is confusing in both its contributions and its technical novelty. I don’t recommend to accept this paper, at least in the current format.

The paper states two major contributions (the last paragraph of Introduction), one is the new model Scratchpad Encoder, and the other is “possible to generate a large high quality (SPARQL query, local form) dataset”. For the second contribution, there isn’t any evaluation/justification about the quality of the generated questions and how useful this dataset would be in any KB-QA applications. I believe that this paper is not the first one to study question generation from logical form (cf. Guo et al, 2018 as cited), so it is unclear what is the contribution of this paper in that respect.

For the modeling contribution, although it shows some improvements on the benchmarks and some nice analysis, the paper really doesn’t explain well the intuition of this “write” operation/Scratchpad (also the improvement of Scratchpad vs coverage is relatively limited). Is this something tailored to question generation? Why does it expect to improve on the question generation or it can improve any tasks which build on top of seq2seq+att framework (e.g., machine translation, summarization -- if some results can be shown on the most competitive benchmarks, that would be much more convincing)?

In general I find Section 3 pretty difficult to follow. What does “keeping notes” mean? It seems that the goal of this model is to keep updating the encoder hidden vectors (h_0, .., h_T) instead of fixing them at the decoder stage. I think it is necessary to make it clearer how s_{post_read} and attn_copy are computed with the updated {h^i_t} and what u^i is expected to encode. \alpha^i_t and u^i are also pretty complex and it would be good to conduct some ablation analysis.

Minor points:
- tau Yih et al, 2016 --> Yih et al, 2016
- It is unclear why the results on WikiSQL is presented in Appendix. Combining the results on both datasets in the experiments section would be more convincing.
- Table 1: Not sure why there is only one model that employs beam search (with beam size = 2) among all the comparisons. It looks strange.

---

> ### Author Response · Authors · 2018-12-11
> **Our contribution is a conceptually simple method achieving state of the art performance**
>
> Your interpretation of section 3 is exactly right. Thank you for suggesting additional experiments to better understand the behavior of the scratchpad component. We would like to note that beyond the gains across all evaluated quantitative metrics (bleu, rouge, meteor), our method shows substantial gains on human evaluations. In future work we propose to use our method to generate a large dataset and evaluate its performance.
> We don’t claim to be the first to generate questions from logical form, but the experiments within show that our approach is superior to standard approaches in the literature.

---

### Official Review · AnonReviewer3 · 2018-11-05
**Interesting idea but not novel enough**

**Rating:** 4
**Confidence:** 4

**Review:**

Overall:
This paper introduces the Scratchpad Encoder, a novel addition to the sequence to sequence (seq2seq) framework and explore its effectiveness in generating natural language questions from a given logical form. The proposed model enables the decoder at each time step to modify all the encoder outputs, thus using the encoder as a “scratchpad” memory to keep track of what has been generated so far and to guide future generation.

Quality and Clarity:
-- The paper is well-written and easy to read.
-- Consider using a standard fonts for the equations.


Originality :
The idea of question generation: using logical form to generate meaningful questions for argumenting data of QA tasks is really interesting and useful.
Compared to several baselines with a fixed encoder, the proposed model allows the decoder to attentively write “decoding information” to the “encoder” output. The overall idea and motivation looks very similar to the coverage-enhanced models where the decoder also actively “writes” a message (“coverage”) to the encoder's hidden states.
In the original coverage paper (Tu et.al, 2016), they also proposed a “neural network based coverage model” where they used a general neural network output to encode attention history, although this paper works differently where it directly updates the encoder hidden states with an update vector from the decoder. However, the modification is slightly marginal but seems quite effective. It is better to explain the major difference and the motivation of updating the hidden states.

-------------------
Comments:
-- In Equation (13), is there an activation function between W1 and W2?
-- Based on Table 1, why did not evaluate the proposed model with beam-search?

---

> ### Author Response · Authors · 2018-12-11
> **Our method is simpler and delivers state of the art performance gains while being conceptually interesting**
>
> We thank the reviewer for their comments and for noting correctly that our modification is quite effective, particularly regarding the large improvements on human evaluations. Our method is simpler in both conception and implementation than coverage, while requiring less parameters and being twice as likely to be chosen as better by human judges. We agree with the reviewer on the simplicity of our method, which we believe to be an asset. In addition to that, we believe the Scratchpad Encoder is fundamentally interesting as a mirror to the ‘attentive read’ common in seq2seq models. We also appreciate the reviewer taking their time to draw our attention to how to better emphasize the novelty and simplicity of our work.

---

### Meta-Review · Area_Chair1 · 2018-12-13
**Not enough novelty relative to neural coverage models**

**Confidence:** 5
**Recommendation:** Reject

**Metareview:**

This paper introduces a "scratchpad" extension to seq2seq models whereby the encoder outputs, typically "read-only" during decoding, are editable by the decoder. In practice, this bears quite a lot of similarity—if not in the general concept, then in the the implementation—to a variety of models proposed in the NLP community (see reviews for details). As the technical novelty of the paper is quite limited, and there are issues with the clarity both in the technical contribution and in presenting what exactly is the main contribution of the paper, I must concur with the reviewers and recommend rejection.